# Estradiol Valerate Affects Hematological and Hemorheological Parameters in Rats

**DOI:** 10.3390/metabo12070602

**Published:** 2022-06-28

**Authors:** Barbara Barath, Adam Varga, Adam Attila Matrai, Krisztina Deak-Pocsai, Norbert Nemeth, Adam Deak

**Affiliations:** 1Department of Operative Techniques and Surgical Research, Faculty of Medicine, University of Debrecen, H-4032 Debrecen, Hungary; barath.barbara@med.unideb.hu (B.B.); varga.adam@med.unideb.hu (A.V.); matrai.adam@med.unideb.hu (A.A.M.); nemeth@med.unideb.hu (N.N.); 2Doctoral School of Clinical Medicine, University of Debrecen, H-4032 Debrecen, Hungary; 3Department of Physiology, Faculty of Medicine, University of Debrecen, H-4032 Debrecen, Hungary; deak-pocsai.krisztina@med.unideb.hu

**Keywords:** polycystic ovary syndrome, animal model, hematology, hemorheology, red blood cell aggregation, red blood cell deformability, glucose, hormones

## Abstract

Polycystic ovary syndrome (PCOS) is one of the most common endocrinological diseases in women. Although the risk of cardiovascular diseases is high in PCOS, the number of scientific publications describing hemorheological changes is not significant. We aimed to perform a comprehensive hematological and micro-rheological study on experimentally induced PCOS in rats.Wistar rats were divided into control (n = 9) and PCOS groups (n = 9), in which animals received single-dose estradiol valerate. Measurements were carried out before treatment and monthly for four months. Bodyweight, blood glucose concentration, hematological parameters, red blood cell (RBC) deformability, and aggregation were measured. A histological examination of the ovary was performed at the end of the experiment. The blood glucose level and the bodyweight were significantly elevated vs. base in the PCOS group. A significant decrease was seen in RBC count, hemoglobin, and hematocrit. The maximal elongation index showed a significant increase. PCOS also resulted in a significant increase in RBC aggregation index parameters. The histological and hormone examinations confirmed developed PCOS. The administration of estradiol valerate caused significant changes during the examined period in hematological and hemorheological parameters. Our results draw attention to the possible usefulness of micro-rheological investigations in further studies on PCOS.

## 1. Introduction

Polycystic ovary syndrome (PCOS) is one of the most common endocrinological diseases among women of childbearing potential [1,2,3]. Its prevalence averages 8–13% [1,2,4]. It has various symptoms, like ovarian dysfunction, induced by the disturbed secretion of gonadotropin-releasing hormone (GnRH), thus the levels of luteinizing hormone (LH) and follicle-stimulating hormone (FSH) are insufficient. There are elevated LH and testosterone levels in women with PCOS, while the FSH levels are in a normal range [3]. The lack of hormonal production can lead to irregular menstruation or amenorrhea. The hormonal imbalance leads to the formation of numerous cysts in the antral follicles, and this phenomenon is defined as polycystic ovary syndrome. Some cysts can produce androgens, which cause virilization, hyperandrogenism, or hyperandrogenemia. In many cases, insulin resistance, dyslipidemia, obesity, hyperinsulinemia, metabolic syndrome, type 2 diabetes, high blood pressure, and cardiovascular diseases (CVD) can occur [1,2,3,5]. Several CVD markers, predictors, and risk factors can be elevated or presented in PCOS, like endothelial dysfunction, C-reactive protein, and coronary artery calcium scores [6].

PCOS is diagnosed according to the Rotterdam criteria that at least two of the following three factors are required to diagnose PCOS if other endocrine diseases can be ruled out: (1) oligomenorrhea or total amenorrhea, (2) signs of androgenic excess (e.g., alopecia), (3) polycystic ovaries [1,3,7]. Four phenotypic manifestations are known. The first, classical phenotype contains all three Rotterdam criteria, the second includes hyperandrogenism and oligo- or amenorrhea, the third consists of androgen excess, and polycystic ovaries, and the fourth with oligo- or amenorrhea and polycystic ovaries. The therapeutic goals are to restore the regularity of the cycle, reduce the symptoms of hyperandrogenism, settle the carbohydrate household, and treat any infertility that may occur. PCOS is currently incurable and requires lifelong treatment to eliminate the symptoms. Daily treatments include oral contraceptives, progesterone therapy, clomiphene citrate, letrozole, metformin, and gonadotropins [4].

The hematological and hemorheological parameters show differences depending on gender, hormonal status, and menstrual cycle [8,9,10]. Guillet et al. found that during menstruation, the red blood cell deformability is decreased in the pre-ovulation and post-ovulation phases [9]. The estradiol concentration has a positive correlation with the whole blood viscosity, as it may cause decreased deformability with increasing packed-cell volume [11]. Hormonal contraceptives affect red blood cell deformability. Grau et al. reported that women with no contraceptive therapy have higher erythrocyte deformability than men, while women on hormonal contraceptives have lower deformability values and moderately increased whole blood viscosity and red blood cell aggregation [10,11].

Although the risk of cardiovascular diseases is high in PCOS, the number of scientific publications describing hemorheological changes is insignificant [12,13]. In a previous study, an increased plasma and whole blood viscosity and significantly increased erythrocyte aggregation in human patients suffering from PCOS were reported; no significant differences in erythrocyte deformability were detected versus the control group [14]. The aforementioned hematological and hemorheological changes have been previously described in rats as well [8,9,15]. The gonadectomy affects hormone homeostasis and consequently the hemorheological and hematological parameters. After gonadectomy, the white blood cell count was elevated in female rats and the red blood cell count was decreased in male rats [10]. The oral contraceptives elevate the body weight, the hematocrit, plasma viscosity, albumin, and total protein levels in rats [11].

Several methods to study PCOS in experimental animals are known. In rats, PCOS can be triggered by using constant light exposure or chronic cold stress [16,17]. Transgenic animals are also available [16]. The most commonly used methods are hormone-generated models, such as androgen-induced PCOS models like testosterone [18], DHEA (Dehydroepiandrosterone) [19,20,21], DHT (5α-dihydrotestosterone) [22,23]; aromatase inhibitor-induced models like letrozole treatment [24,25]; and progesterone receptor antagonist- or estrogen-induced models like the estradiol valerate treatment [24,26,27].

Administration of single-dose estradiol valerate is an easy, feasible model, without repetitive interventions on the animals. In this model, development of the PCOS must be confirmed when investigating any other parameters. However, no data were found for the micro-rheological examination of experimental animal models of PCOS in the literature. We hypothesized that PCOS causes impairment of red blood cell deformability and aggregation. Our aim was, thus, to perform a comprehensive hematological and micro-rheological study on experimentally induced PCOS in rats.

## 2. Results

### 2.1. Bodyweight and General Observations

Examining the weight of Control animals, we observed a significant increase versus Base data (*p* = 0.017) in relative values on Day 120. An absolute (Day 120 vs. Base *p* = 0.003) and relative (Day 120 vs. Base *p* = 0.03) increase in body weight was detected in the PCOS group (Figure 1).

No change in the animals’ behavior was observed. Their fur was tidy, their eyes were clear, and no porphyrin discoloration was visible. We saw no difference in food and water intake between the two groups.

### 2.2. Vaginal Smears

Examining the vaginal smears, we found that the estrus cycle of the PCOS group has stopped in the pro-estrous (77.78%) and met-estrous (22.22%) phase from the 90-day mark (Day 90) of the experiment and had not been recovered, while the estrus cycle of the Controls remained normal (Table 1).

### 2.3. Hormone Concentrations

In the FSH levels, we did not find significant differences in either group. The LH values increased significantly in the PCOS group in the fourth month compared to the base values (*p* = 0.036). The testosterone levels were elevated in both groups (Table 2).

### 2.4. Fasting Glucose and Oral Glucose Tolerance Test Results

Blood glucose levels (Figure 2A) rose steadily and by the fourth month was significantly higher than baseline (*p* = 0.021). During the oral glucose tolerance test (OGTT), the Control group and the PCOS group had a significant increase in blood glucose by 15 min compared to baseline (Figure 2B). In the PCOS group, an increase in blood glucose until 30 min and then a steep decrease was observed. The values of the Control group showed a decrease after a peak measured at 15 min. For both groups, we obtained significantly higher results at all time points than for the baseline measurements; however, there was no statistically significant difference between the two groups (Figure 2B).

### 2.5. Histology

The ovaries in the PCOS group showed macroscopic characteristics: large ovaries with cystic follicles visible through the surface layer, and the uterus was enlarged and edematous. Histopathological examination of ovaries showed polycystic ovarian tissue in the PCOS group, while the Control group showed normal ovarian structure. In the PCOS group, many cystic dilating follicles, a decreased number of corpora lutea, and a significantly decreased number of follicles were present (Table 3). Ovaries in the Control group showed multiple primary, secondary, and antral follicles (Figure 3).

### 2.6. Hematological and Hemorheological Parameters

Hematological parameters are summarized in Table 4.

A significant decrease vs. baseline (Base) and vs. Control group in red blood cell count (Day 120 vs. Base *p* = 0.015, Day 120 vs. Control *p* < 0.001), the hematocrit (Day 120 vs. Base *p* = 0.004, Day 120 vs. Control *p* = 0.004), and hemoglobin values (Day 60 vs. Base *p* < 0.001, Day 60 vs. Control *p* < 0.001, Day 120 vs. Control *p* < 0.001) in the PCOS group were detected. In the mean corpuscular volume (MCV; Day 90 vs. Control *p* = 0.006, Day 30 vs. Base *p* = 0.015, vs. Control *p*= 0.017) and the mean corpuscular hemoglobin content (MCH; Day 90 vs. Base *p* < 0.001, vs. Control *p* = 0.006, Day 120 vs. Base *p* = 0.002; Day 30 vs. Control *p* = 0.02), a significant increase was observed in the PCOS group after the initial decrease. There was a continuous increase in the mean corpuscular hemoglobin concentration (MCHC) values in the PCOS group (Day 60 vs. Base *p* < 0.001, Day 90 vs. Base *p* = 0.003, Day 120 vs. Base *p* = 0.001).

In the PCOS group, there was a significant decrease in the white blood cell count (Day 120 vs. Base *p* = 0.03, Day 90 vs. Control *p* = 0.041) and a significant increase in the platelet count (Day 120 vs. Base *p* < 0.001).

The RBC deformability describing parameter (the elongation index (EI) at a shear stress of 3 Pa) values were increased in the PCOS group at the 60th (vs. Base *p* = 0.047) and 90th days (vs. Base *p* < 0.001). For the maximal elongation index (EI_max_), a significant increase was observed in the PCOS group (Day 120 vs. Base *p* = 0.035). The SS_1/2_ values (shear stress at half EI_max_) show a significant decrease at the 90th (vs. Base *p* < 0.001, vs. Control *p* < 0.001) and 120th days (vs. Base *p* < 0.001). On the 90th (vs. Base *p* < 0.001) and 120th day (vs. Base *p* < 0.001) tests the EI_max_/SS_1/2_ ratio was significantly elevated in the PCOS group (Table 5).

In the aggregation index in stasis at 5 s (M 5s; Day 120 vs. Base *p* < 0.001, vs. Control *p* = 0.004) and the aggregation index at 3 s^−1^ shear stress at 5 s (M1 5s; Day 120 vs. Base *p* < 0.001, vs. Control *p* = 0.015) (Figure 4A,B) in the PCOS group, there was a significant, continuous increase. In the aggregation index in stasis at 10 s (M 10s; Day 120 vs. Base *p* = 0.005) and aggregation index at 3 s^−1^ shear stress at 10 s (M1 10s; Day 120 vs. Base *p* < 0.001) (Figure 4C,D), there was also an increase in the first two months, but after, it slightly decreased.

## 3. Discussion

The etiology of polycystic ovary syndrome is not clearly described, but it is well-known that PCOS is one of the most frequent endocrinological diseases in women of reproductive age [1,2]. PCOS has various effects on blood circulation. PCOS causes raised plasma viscosity, which affects microcirculation [28]. Besides that, PCOS is characterized by increased testosterone, luteinizing hormone, fasting insulin levels, higher triglyceride, and fibrinogen concentration. The fibrinogen level has a positive correlation with plasma viscosity [29]. The elevated plasma viscosity increases the whole blood viscosity, and this can impair the blood flow [14]. Simmonds et al. examined the hemorheological changes between age-matched healthy volunteers and women who suffered from PCOS [14]. They describe enhanced red blood cell aggregation, increased whole blood and plasma viscosity, decreased hematocrit, and unchanged red blood cell deformability in the PCOS group [14]. The main goals of therapy are to recover the regularity of the menstrual cycle, treat infertility, and stabilize the carbohydrate household. Infertility treatments include oral contraceptives, progesterone therapy, and clomiphene citrate [4]. For insulin resistance and type 2 diabetes, the most frequent treatment is metformin [4]. In recent years, various inositol derivatives, which have a key role in the insulin signaling pathway as second messengers, have become a common medication for treating insulin resistance. In addition, inositol isoforms affect ovulation, thereby may help to treat infertility [30,31].

There are controversial data on the effect of estradiol valerate (EV) on glucose metabolism. Elevated fasting glucose levels [24,32], or no changes between the treated and control groups were reported. Dănăsea et al. described elevated fasting glucose levels, but no differences in the oral glucose tolerance test [24]. We found constantly rising fasting glucose levels in the PCOS group, and we could not observe any significant differences between the two groups in the OGTT measurements [27]. To induce PCOS, we used a single dose of estradiol valerate [24,26]. The estradiol valerate is a long-acting estrogen that causes the dysregulation of GnRH on the hypothalamic–pituitary axis, resulting in incorrect release and/or storage of LH [27,33]. In the EV-induced model, the effect on testosterone levels depends on the dosage of the estradiol valerate. Using 2 mg EV, the testosterone level increases [16,24,27,32,33,34]; with administration of 4 mg EV, the free testosterone levels mostly decrease in rats, although elevated testosterone levels with this dosage were reported as well [16,24,35]. In our study, the testosterone levels were elevated in both groups, but there were no significant differences either within groups or between the Control and PCOS groups. Controversial data regarding the effect of EV treatment on LH levels were reported. Several studies reported that LH levels increased in the EV-induced model [27,34,35], while others found decreasing LH levels [16,27,34]. Usually, a reduction or no change can be detected in the FSH levels in the estrogen-generated models [15,26,30]. In our investigation, the LH levels were elevated in the PCOS group, while in the FSH levels, we did not find differences in either group.

As a result of the EV treatment, the estrous cycle of the rats is disturbed and stopped. The human menstrual cycle has two phases: the follicular phase and the luteal phase. The follicular phase starts on the first day of menstruation and holds up to the ovulation (1–14 days); the luteal phase begins after ovulation and stops 14 days after (15–28 days). The estrous cycle in rats has four phases; these are pro-estrous, estrous, met-estrous, and di-estrous [36,37,38]. Lara et al., established that the estrous cycle of the rats stopped in di-estrous [36]. We found that the PCOS group’s estrous cycle stopped in the pro-estrous (77.78%) and met-estrous (22.22%) phases. The cycle of the control animals was normal. As the Lee–Boot effect (without the presence of males, the cycle of female rodents stops or slows down [39]) can be ruled out, the acyclicity can be attributed to hormone treatment.

Fertility in women decreases with age, caused by changes in ovarian function, a decrease in the quantity and quality of ovaries, and hormonal changes [40]. This ovarian aging is also observed in rats. Anzalone et al., reported that there are fewer follicles in middle-aged rats than in young animals [41]. In the case of this model, the number of cystic follicles and primary follicles increases in the ovaries [16,27,32,33,42], and no new corpora lutea are found [16,32,43,44], being replaced by regressive old corpora lutea [42]. In our study, there were many cystic follicles and a decreased number of corpora lutea and follicles in the PCOS group. Kang et al., in their model, used a continuous light environment and observed enlarged ovaries and visible, large cysts on the surface of the ovaries [17]. Mirabolghasemi et al., examined the changes in the uterine tissue, and they found that the luminal epithelium height and the thickness of the uterine wall in the PCOS group were increased [26]. We observed that, macroscopically, the uterus was enlarged and edematous, and the ovaries were large and had cystic follicles visible through the surface layer.

It has been previously described that there are sex differences in the hemorheological parameters not only in humans, but in laboratory animals as well. In males, we can find higher aggregation indices and lower deformability results [45,46]. In female rats, there were larger aggregation and deformability results than in male rats [15]. Nemeth et al. reported that after gonadectomy, the deformability slightly decreased both in female and male rats [10]. The values of hemorheological parameters are influenced by the menstrual cycle due to constantly changing hormone levels [47,48]. Various studies described that the sex-hormones influence the rheological parameters, but the mechanism is not completely elucidated [49,50]. Estradiol can have a defensive effect on ischemic damage but elevates the risk of thrombotic events [51,52]. Hemorheological studies in women during menopause have shown a reduction in whole blood viscosity and red blood cell aggregation; however, an increase in red blood cell deformability was detected [53].

In our experiment, as a result of estradiol-valerate treatment, we established a decrease in red blood cell count, hemoglobin, and hematocrit levels, and an increase in aggregation indices, since the first month of research. A significant increase in EI at 3 Pa and EI_max_ values was observed in the PCOS group; this change may be due to the administration of a large amount of estrogen. In an in vitro experiment, Farber et al. showed that exogenous estrogen increases the deformability of red blood cells, but they did not find any differences in the red blood cell aggregation [27]; however, Brun et al. reported that estradiol decreases deformability and increases eryptosis [44]. In the white blood cell count, we observed a significant decrease in the PCOS group, while the Controls’ values decreased also, but not to the same extent. The aging of the animals may also play a role in these changes. The number of white blood cells and the elongation index in female rats decreases with aging, while the red blood cell aggregation index parameters show a slight increase [8,9].

As all scientific studies have their shortcomings, our experiment has its limitations as well. One of these limitations is the difference between the duration of the rats’ estrous cycle and women’s menstrual cycle; since the rats’ cycle is shorter (4 days), the changes in hormone levels are faster. Another limiting factor is that our experimental model is dose-dependent, thus the administration of estradiol valerate may not, in all experiments, lead to typical androgen change, weight gain (obesity), and insulin resistance.

## 4. Materials and Methods

The study was carried out according to the EU regulation (EU Directive 63/2010) and The Hungarian Animal Protection Act (Law XXVIII/1998) on animal experimentation. The study was registered by the University of Debrecen Committee of Animal Welfare (permission registration No.: 17/2019/UDCAW). The animals were housed in a conventional animal facility, with a 12-h light/dark cycle, at a temperature range between 20 °C to 24 °C. Standard rodent nutrition pellets and tap water were available ad libitum.

### 4.1. Bodyweight Measurements and General Observations

The body weights of the animals were monitored weekly, and the behavior and physical condition of the animals were observed.

### 4.2. Determining the Estrous Cycle

After ten days of acclimatization time, 18 adult female Crl: WI rats were included in the experiment. Before the start of the experiment, the animals were checked for the normal estrous cycle by vaginal smear sampling, and after it, they were divided into two groups (n = 9/group). The control group (Control) did not receive any treatment. The PCOS group animals received 4 mg estradiol valerate (estradiol valerate, Y0000046, Sigma-Aldrich, St. Louis, MO, USA) per animal, dissolved in 0.5 mL sesame oil (sesame oil, S3547, Sigma-Aldrich, St. Louis, MO, USA), as a subcutaneous injection.

Vaginal smears were taken on five consecutive days each month using the lavage technique [34,35] with 250–500 µL distilled water, filtered pipette tips, and an eyedropper. The smears were dried overnight at room temperature and then stained by the modified Giemsa staining described previously [8]. Smears were fixed in methanol (Methanol Technical, 20903, VWR International, Radnor, PA, USA) for half a minute and then stained using Giemsa dye (PanReac AppliChem Giemsa’s Azur-Eosin-Methylene Blue solution, 251338, ITW Reagents) for 1 min, then washed with distilled water and dried overnight at room temperature. The evaluation was performed by light microscopy (Nikon Eclipse E200, Nikon Europe BV, Amsterdam, The Netherlands).

### 4.3. Blood Sampling Method

Blood samples were taken from the lateral caudal vein into vacuum tubes containing K3-EDTA as an anticoagulant (BD Vacutainer^®^ Becton, Dickinson and Company, Franklin Lakes, NJ, USA), before the treatment, and then monthly for four months thereafter, under anesthesia using a mixture of ketamine (100 mg/kg, CP-Ketamin—ketamine hydrochloride 10%; Produlab Pharma BV, Raamsdonksveer, The Netherlands), xylazine (10 mg/kg, CP-Xylazin—xylazine-hydrochloride, 2%; Produlab Pharma BV, Raamsdonksveer, The Netherlands), and atropine (0.05 mg/kg, atropinum sulfuricum 0.1%, Egis Pharmaceuticals PLC, Budapest, Hungary). We used a 26G cannula (Becton, Dickinson and Company, Franklin Lakes, NJ, USA), and a maximum of 750 μL of blood was collected at each time point per animal.

### 4.4. Laboratory Measurements

To confirm the development of PCOS, hormone measurements were performed via the ELISA method (LH, FSH, testosterone, IBL International GmbH, Hamburg, Germany). The LH and FSH values were measured with solid-phase sandwich ELISA, the wells were coated with monoclonal anti-LH or anti-FSH antibodies, and samples were incubated with enzyme-conjugated horseradish peroxidase. After 30 min, we washed off the unbound conjugates and gave the substrate solution to the reaction. The testosterone measurements were performed with a competition-based method. The samples with unknown testosterone concentration and a fixed amount of enzyme-labeled antigen compete for the binding sites of the antibodies coated onto the wells. After 60 min, the plate was washed to stop the competition reaction; then, we added the substrate solution. At the end of both methods, the intensity of the colors could be measured with an ELISA reader, and using a standard curve, we could calculate the hormone concentration of the samples.

Blood glucose levels were monitored weekly by a hand-held glucometer (AccuCheck Active glucometer, Roche Diabetes Care GmbH, Mannheim, Germany). On the 120th day of the experiment, we performed an oral glucose tolerant test (OGTT) as follows: after 12 h of fasting, 2 g/kg of glucose [54,55] was dissolved in 1.5 mL of distilled water and was administered by oral gavage. The blood glucose levels (mmol/L) were measured at six different time points (before and 15, 30, 60, 90, 120 min after the glucose administration).

Quantitative and qualitative hematological parameters (white blood cell count, red blood cell count, hemoglobin, hematocrit, mean cell volume [MCV], mean cell hemoglobin [MCH], mean corpuscular hemoglobin concentration [MCHC], platelet count) were determined by the Sysmex K-4500 hematological analyzer (TOA Medical Electronics Co., Ltd., Kobe, Japan). The device determines the white blood cell, red blood cell, and platelet count with aperture-impedance and the hemoglobin concentration by spectrophotometry. The other parameters were calculated from this data.

The deformability of the erythrocyte is the ability of the red blood cells to passively deform under the influence of force. The deformability was measured in a high-viscosity medium (Polyvinylpyrrolidone [PVP] solution, pH: 7.0–7.2, viscosity:25.2–28.0 mPa s) using LoRRca MaxSis Osmoscan ektacytometer (Mechatronics BV, Zwaag, The Netherlands). For the measurements, 10 μL of blood was suspended in 2 mL of PVP solution. During the analysis, the sample was exposed to 0.3–30 Pa shear stress. The elongation indices were calculated from the diffraction patterns taken at each shear stress value [56,57].

We determined the red blood cell aggregation, which is defined as the reversible association of erythrocytes on a sufficiently low shear rate or in the presence of stasis, with the Myrenne MA-1 aggregometer, (Myrenne GmbH, Roetgen, Germany). This aggregometer uses a red-blood-cell-aggregation measuring method based on light transmission. The 20 μL blood sample was applied to a two-glass lens and spread on the surface of the lens with a glass plate. The instrument disaggregates the sample and then measures the intensity of the transmitted light in seconds 5 and 10 of the process. The detector is located under the infrared light source above the lens [56].

The hematological and hemorheological measurements were carried out within 4 h.

### 4.5. Histological Examination

The histological examination of the ovaries was performed at the end of the experiment. The animals were sacrificed with an injection of a triple dose of anesthetic (300 mg/kg, ketamine hydrochloride 10%; 30 mg/kg, xylazine-hydrochloride). Ovaries were fixed in 10% neutral formaldehyde (Leica Biosystems Inc., Leider Lane Buffalo Grove, IL, USA) and then embedded in paraffin. Then, 6 µm slices were taken and dyed with hematoxylin-eosin (Sigma-Aldrich, St. Louis, MO, USA). The number of follicles, corpora lutea, and cystic follicles was recorded in the ovarian tissue of each rat at low magnification.

### 4.6. Statistical Analysis

For statistical analysis, paired *t*-test, Wilcoxon test, Student’s *t*-test, and Mann–Whitney rank-sum test were used, depending on the distribution of the data, with SigmaStat (Systat Software Inc., San Jose, CA, USA) software. Probabilities less than *p* = 0.05 were considered significant.

## 5. Conclusions

In our study, PCOS-specific alterations were detected in the metabolic parameters (increase in body weight and elevated fasting blood glucose levels), in the estrous cycle (disrupted estrous cycle), and histological features (lack of mature follicles and presence of large cysts). The administration of hormone caused significant changes during the examined period in hematological (a decrease in red blood cell count, hematocrit, and hemoglobin values), and hemorheological parameters. To the best of our knowledge, no one has previously described red blood cell aggregation and erythrocyte deformability increase during estradiol-valerate-induced PCOS in rats. Our results draw attention to the possible usefulness of micro-rheological investigations in further studies on PCOS.

## Figures and Tables

**Figure 1 metabolites-12-00602-f001:**
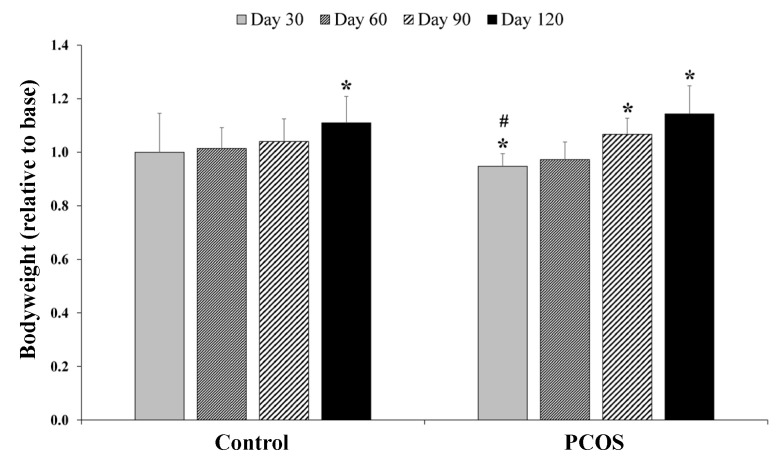
Changes in the bodyweight of the Control and polycystic ovary syndrome (PCOS) groups. Means ± S.D.; * *p* < 0.05 vs. Base, # *p* < 0.05 vs. Control.

**Figure 2 metabolites-12-00602-f002:**
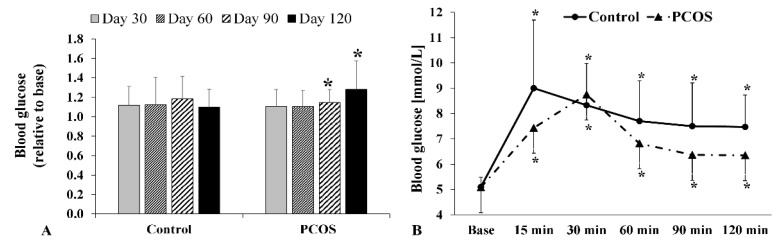
Changes in blood glucose concentration (**A**) and the results of oral glucose tolerance test (OGTT) (**B**) of the Control and polycystic ovary syndrome (PCOS) groups. Means ± S.D.; * *p* < 0.05 vs. Base.

**Figure 3 metabolites-12-00602-f003:**
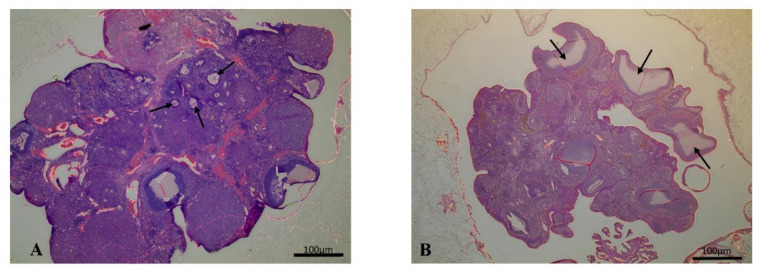
Representative pictures of histological appearance of the normal ovary (**A**) and the polycystic ovary (**B**). H&E staining, original magnification: 4×. Arrows show the cysts.

**Figure 4 metabolites-12-00602-f004:**
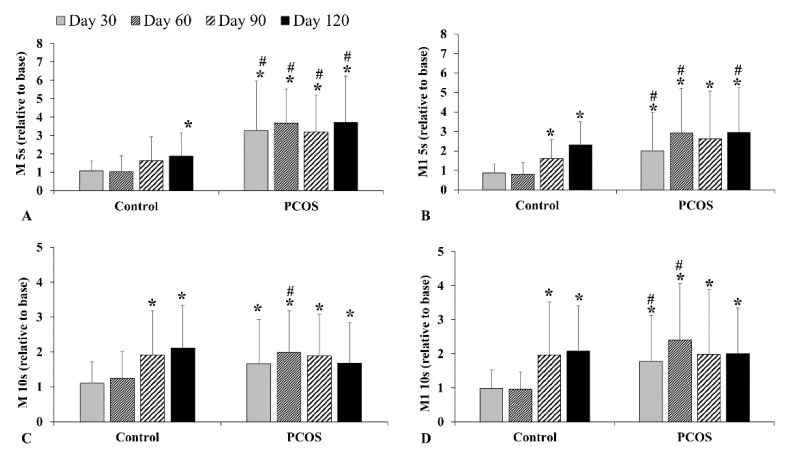
Alteration of red blood cell aggregation index parameters, such as M 5s (**A**), M1 5s (**B**), M 10s (**C**), and M1 10s (**D**). Means ± S.D.; * *p* < 0.05 vs. Base, # *p* < 0.05 vs. Control.

**Table 1 metabolites-12-00602-t001:** Distribution of estrus cycle phases in the animals at the beginning of the experiment (Base), and on Day 30, 60, 90, and 120. The percentage of the rats in each phase was calculated based on the microscopic picture of the vaginal smears.

Phase	Base		Day 30		Day 60		Day 90		Day 120	
	Control	PCOS	Control	PCOS	Control	PCOS	Control	PCOS	Control	PCOS
Pro-estrous	-	33.3%	22.22%	33.33%	44.44%	55.55%	62.5%	77.78%	12.5%	77.78%
Estrous	66.6%	33.3%	55.56%	-	22.22%	11.11%	25%	-	62.5%	-
Met-estrous	33.3%	-	22.22%	44.44%	11.11%	33.33%	12.5%	22.22%	25%	22.22%
Di-estrous	-	33.3%	-	22.22%	22.22%	-	-	-	-	-

**Table 2 metabolites-12-00602-t002:** Relative changes (vs. Base) in the FSH, LH, and testosterone levels in Control and polycystic ovary syndrome (PCOS) group.

Hormone	Group	Day 60	Day 120
LH	Control	0.996 ± 0.021	0.998 ± 0.029
	PCOS	1.00 ± 0.012	1.008 ± 0.022 ^+^
FSH	Control	0.986 ± 0.039	0.988 ± 0.056
	PCOS	0.985 ± 0.037	1.018 ± 0.062
Testosterone	Control	1.376 ± 0.292	1.533 ± 0.677
	PCOS	1.206 ± 1.032	1.472 ± 1.000

Means ± S.D.; ^+^ *p* < 0.05 vs. Day 60.

**Table 3 metabolites-12-00602-t003:** Number of follicles, corpus lutea, and cysts in the Control and PCOS groups.

Structure or Variable	Control	PCOS
Follicle	6.71 ± 6.01	1.5 ± 1.0
Corpus Luteum	2.83 ± 2.13	1.63 ± 0.9
Cyst	0	5 ± 4.42 *
Cyst diameter (μm)	-	690 ± 92.14 *

Means ± S.D.; * *p* < 0.05 vs. Control.

**Table 4 metabolites-12-00602-t004:** Changes in the hematological parameters in Control and polycystic ovary syndrome (PCOS) group.

Variable	Group	Base	Day 30	Day 60	Day 90	Day 120
WBC [10^9^/L]	Control	4.37 ± 0.83	5.27 ± 0.87	5.11 ± 1.61	5.41 ± 1.73	4.61 ± 1.1
	PCOS	5.25 ± 1.05 ^#^	5.1.4	4.8 ± 1.48	3.97 ± 0.97 *^,#^	3.96 ± 1.3 *
RBC [10^12^/L]	Control	7.12 ± 0.71	6.86 ± 0.42	7.38 ± 0.49	7.42 ± 0.38 *	7.48 ± 0.32 *
	PCOS	7.33 ± 0.34	6.68 ± 0.3 *	6.41 ± 0.55 *^,#^	6.8 ± 0.39 *^,#^	6.85 ± 0.54 *^,#^
Hct [%]	Control	41.56 ± 4.2	40.24 ± 2.6	43.01 ± 2.48	43.11 ± 1.96	44 ± 2 *
	PCOS	42.89 ± 1.82	38.37 ± 1.74 *^,#^	37.34 ± 3.89 *^,#^	40.12 ± 2.17 *^,#^	40.17 ± 2.56 *^,#^
Hbg [g/dL]	Control	13.8 ± 1.4	13.37 ± 0.94	14.34 ± 0.86	14.29 ± 0.68 *	14.56 ± 0.73 *
	PCOS	14 ± 0.58	12.68 ± 0.61 ^#^	12.39 ± 1.31 ^#^	13.46 ± 0.81 *^,#^	13.47 ± 0.83 *^,#^
MCV [fL]	Control	58.3 ± 0.74	58.6 ± 1.05	58.33 ± 1.22	58.14 ± 1.12	58.85 ± 1.3
	PCOS	58.5 ± 1.17	57.43 ± 1.7 *^,#^	58.1 ± 2.00	58.94 ± 0.91 ^#^	58.75 ± 1.71
MCH [pg]	Control	19.4 ± 0.52	19.5 ± 0.61	19.45 ± 0.65	19.28 ± 0.56	19.48 ± 0.57
	PCOS	19.11 ± 0.45	18.93 ± 0.69 ^#^	19.27 ± 0.63	19.8 ± 0.67 *^,#^	19.71 ± 0.66 *
MCHC [g/L]	Control	33.2 ± 0.82	33.2 ± 0.68	33.34 ± 0.75	33.14 ± 0.44	33.09 ± 0.39
	PCOS	32.66 ± 0.41 ^#^	32.97 ± 0.48	33.18 ± 0.44 *	33.54 ± 0.96 *	33.56 ± 0.68 *
Plt [10^9^/L]	Control	608.6 ± 120.7	597.33 ± 213.45	659 ± 118.9	559.9 ± 195.3	600.6 ± 185.4
	PCOS	534.29 ± 87.69 ^#^	665.2 ± 204.4 *	595.94 ± 136.97	578.67 ± 145.7	645.53 ± 92.57 *

Means ± S.D.; * *p* < 0.05 vs. Base, ^#^ *p* < 0.05 vs. Control, WBC: white blood cell count; RBC: red blood cell count; Hct: hematocrit; Hbg: hemoglobin concentration; MCV: mean corpuscular volume; MCH: mean corpuscular hemoglobin; MCHC: mean corpuscular hemoglobin concentration; Plt: platelet count.

**Table 5 metabolites-12-00602-t005:** Changes in the parameters describing the red blood cell deformability in Control and polycystic ovary syndrome (PCOS) group.

Variable	Group	Base	Day 30	Day 60	Day 90	Day 120
EI at 3 Pa	Control	0.38 ± 0.01	0.38 ± 0.02	0.4 ± 0.01 *	0.38 ± 0.01	0.39 ± 0.01 *
	PCOS	0.38 ± 0.02	0.38 ± 0.01	0.39 ± 0.01 *	0.39 ± 0.01 *^,#^	0.39 ± 0.01
EI_max_	Control	0.55 ± 0.02	0.56 ± 0.02	0.57 ± 0.02 *	0.56 ± 0.03	0.57 ± 0.03 *
	PCOS	0.54 ± 0.02	0.57 ± 0.02 *	0.57 ± 0.01 *	0.56 ± 0.03 *	0.56 ± 0.24 *
SS_1/2_ [Pa]	Control	1.78 ± 0.22	1.79 ± 0.3	1.73 ± 0.22	1.73 ± 0.19	1.14 ± 0.33 *
	PCOS	1.62 ± 0.2 ^#^	1.75 ± 0.2 *	1.78 ± 0.18 *	1.16 ± 0.34 *^,#^	1.24 ± 0.24 *
EI_max_/SS_1/2_ [Pa^−1^]	Control	0.31 ± 0.04	0.33 ± 0.08	0.33 ± 0.05	0.32 ± 0.03	0.49 ± 0.11 *
	PCOS	0.34 ± 0.04	0.33 ± 0.04	0.33 ± 0.04	0.47 ± 0.1 *^,#^	0.46 ± 0.09 *

Means ± S.D.; * *p* < 0.05 vs. Base, ^#^ *p* < 0.05 vs. Control, EI at 3 Pa: elongation index at 3 Pa shear stress; EI_max_: maximal elongation index; SS_1/2_: shear stress at half EI_max._

## Data Availability

The raw data supporting the conclusions of this article will be made available by the authors, without undue reservation.

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
