# Peer review of "Estradiol Valerate Affects Hematological and Hemorheological Parameters in Rats"

_metabolites, 2022, doi:10.3390/metabo12070602_

Round 1
Reviewer 1 Report
The manuscript presents an original study. The aim was to investigate the hematological and hemorheological changes in an experimental model of polycystic ovary syndrome. The results of the study are clearly presented and the discussion is interesting.
However, some improvements are necessary.
Rows 46-47 Revision is necessary. The endothelial dysfunction cannot be elevated, it is present. C-reactive protein is not a marker for CVD but is a predictor and risk factor for CVD.
60-71 I do not think it is useful to compare the hematological parameters between men and women. Such information may be excluded.
66-67 Revision is necessary: “and it is impaired with deformability”
Table 3 - there is an aspect issue in column 1. The abbreviations must be moved after the table.
Table 4 - The abbreviations must be moved after the table.
Please be constant in format when writing the 30th, 60th, 90th and 120th
MCV, MCH, MCHC, SS ½ and EImax – abbreviations must be explained in the main text, not only in the legend of the tables.
M 5s, M1 5s, M 10s, and M1 10s – must be explained at the first use.
Rows 240-241 It is not clear. Please reformulate. Simmonds et al described expanded red blood cell aggregation, whole blood, and plasma viscosity…
Comments regarding the limitations of the study must be added.
Thank you!
Reviewer 2 Report
Reviewer Comments to Author
In this study, the authors revealed hematological and microrheological changes in rats with experimentally induced polycystic ovary syndrome (PCOS), which is somewhat innovative and meaningful for guiding the research on the pathogenesis of PCOS. However, there are some problems to be further improved as well:
1. Why is Estradiol-Valerate chosen as the study variable rather than the drug required in other animal models? It is best to add relevant content in the section of introduction;
2. How is the animal model of PCOS constructed?
3. In this experiment, the changes of blood hormone concentration, the abnormal results of fasting blood glucose and oral glucose tolerance tests, and the changes of oestrus cycle in rats are all to explain the successful construction of the rat model of PCOS. The focus of this experiment is the effect of estradiol valerate on Hemorheology and micro rheology in rats, but the discussion part does not analyze the results in this regard;
4. Some studies have proved the change of microrheologic in PCOS women. What is the significance of doing similar experiments on rats in this study?
5. Putting result 2.6 before result 2.5 will make the article more logical;
6. The clarity of Figure 1 is not enough to see the vertical coordinates clearly; What does the number "131" in Table 2 represent? The first column of Table 3 is out of order.
Round 2
Reviewer 2 Report
The author has made relevant revisions to the manuscript according to the suggestions, and the manuscript can be accepted after revision and addition of relevant references.